ecology/molecular biology

frog disease, alveolate parasites, quantitative PCR, Perkinsea, NAG01

**Authors for correspondence:**
Vanessa Smilansky
e-mail: v.a.smilansky@exeter.ac.uk
David S. Milner
e-mail: david.milner@zoo.ox.ac.uk

# A novel duplex qPCR assay for stepwise detection of multiple Perkinsea protistan infections of amphibian tissues

Vanessa Smilansky[1], Aurelie Chambouvet[2], Mari Reeves[3], Thomas A. Richards[4] and David S. Milner[4]

[1]Living Systems Institute and Biosciences, University of Exeter, Stocker Road, Exeter, Devon EX4 4QD, UK
[2]CNRS, Univ Brest, IRD, Ifremer, LEMAR, Plouzané, France
[3]US Fish and Wildlife Service, Pacific Islands Fish and Wildlife Office, Honolulu, Hawai'i
[4]Department of Zoology, University of Oxford, 11a Mansfield Road, Oxford OX1 3SZ, UK

VS, 0000-0002-1904-8843; AC, 0000-0003-3932-0098;
TAR, 0000-0002-9692-0973; DSM, 0000-0003-3669-7463

Alveolate protists within the phylum Perkinsea have been found to infect amphibians across a broad taxonomic and geographic range. Phylogenetic analysis has suggested the existence of two clades of amphibian Perkinsea: a putatively non-pathogenic clade linked to asymptomatic infections of tadpoles in Africa, Europe and South America, and a putatively pathogenic clade linked to disease and mass mortality events of tadpoles in North America. Here, we describe the development of a duplex TaqMan qPCR assay to detect and discriminate between rDNA sequences from both clades of Perkinsea in amphibian tissues. The assay uses a single primer pair to target an 18S small subunit (SSU) ribosomal RNA (rRNA) gene region shared between the two clades, and two dual-labelled probes to target a region within this fragment that is diagnostic for each clade. This assay enables rapid screening for each of the two Perkinsea groups, allowing for detection, primarily of the phylogenetic group associated with disease outbreaks, and secondarily for the phylogenetic group with no current disease relationship identified. Incorporation of our novel qPCR assay into the routine surveillance of amphibian populations will allow for the assessment of the incidence of each protist clade, thereby providing an improved understanding of Perkinsea infection pervasiveness and a method to underpin future conservation planning.

# 1. Introduction

Tadpole mass mortality events (MMEs) associated with an alveolate protist in the phylum Perkinsea have been reported across a broad geographic range within the USA [1–5]. Recent phylogenetic analysis of Perkinsea identified these putative pathogens as a distinctive clade within a broad monophyletic group of alveolates known as novel alveolate group 01 (NAG01; [6]). NAG01 includes three clades (NAG01a-c) known to infect tadpoles from a broad range of taxa without causing any identifiable gross or tissue-level symptoms of disease [6], and a discrete clade, designated pathogenic Perkinsea clade (PPC), which is associated with severe Perkinsea infection (SPI) [2]. PPC sequences have been detected in all SPI-associated MMEs described in the USA [1,2], but from only a small proportion of phenotypically normal tadpoles [2], indicating that PPC is the likely causative agent of SPI. Nevertheless, the exact relationship between the PPC clade, the wider NAG01 clade, and SPI-associated disease remains undefined and untested.

Given the additional uncertainty regarding the impact of NAG01a-c infections on tadpole populations [6,7], this led us to develop a method which would allow us to investigate the prevalence of the wider NAG01 group. A quantitative PCR (qPCR) assay specifically for PPC has been developed previously [8]; however, similar methods have yet to be developed for the NAG01a-c clade. Here, we describe the development of a rapid, sensitive, duplex TaqMan qPCR assay to detect and discriminate PPC and NAG01a-c Perkinsea. We also demonstrate that this assay can be performed on handheld qPCR instruments, opening up the possibility for rapid in-the-field testing of NAG01 infections in wild and captive tadpole populations.

# 2. Material and methods

## 2.1. Oligonucleotides

The primers used in this study (table 1) target an approximately 287–290 bp fragment of the 18S rRNA-encoding gene of NAG01 Perkinsea. Specifically, the PPC assay targets this region in a *Rana sphenocephala* pathogen (accession: EF675616) [1], while the NAG01a-c assay targets this region in a randomly selected NAG01-targeted clone (accession: KP122572) [6] (figure 1). The probes used in this study (table 1) target an approximately 113–116 bp variant region within this fragment that can be used to distinguish NAG01a-c from PPC (figure 1). The NAG01R_1 primer and both probes were designed manually, based on the full alignment of Perkinsea sequences available at https://zenodo.org/record/12712#.YEaHkZunyUk. The PPC probe (FAM fluorophore; emission 520 nm) is identical to 93/93 (100%) PPC clone libraries recovered from tadpole liver by Isidoro-Ayza *et al.* [2], and the NAG01a-c probe (Cy5 fluorophore; emission 673 nm) is identical to 172/177 (97%) NAG01a-c clone libraries recovered from tadpole liver by Chambouvet *et al.* [6]. Probe specificity was confirmed using a BLASTn search against the NCBI nr database (electronic supplementary material, table S1), with neither probe having any significant hits (i.e. greater than 75% query sequence coverage) against any frog sequences in this database.

## 2.2. Tissue sample DNA

### 2.2.1. Pathogenic Perkinsea clade assay

Tissue samples were derived from tadpoles collected from three Perkinsea-positive sites in the Kenai Peninsula Borough in Alaska, USA in June/July 2016 (electronic supplementary material, table S3). Liver tissue samples were removed and placed in RNAlater stabilization solution (Invitrogen), transported to the University of Exeter and stored at −80°C. The livers were enlarged and yellow in appearance and were revealed to be heavily infected with Perkinsea organisms upon microscopic observation. Total DNA (KNA_DNA) was extracted from the pooled livers of nine tadpoles (electronic supplementary material, table S3) using the RNeasy PowerSoil DNA Kit (Qiagen). These tadpole livers were small, so were pooled in this instance to ensure sufficient material for the DNA extraction protocol.

### 2.2.2. NAG01a-c assay

Tissue samples were derived from two ethanol-preserved tadpoles collected in 2013 from French Guiana: G2.13 and G8.1 (electronic supplementary material, Table S3). Whole livers were removed and processed for DNA extraction by Chambouvet *et al.* [6]. These samples did not display gross or tissue-level symptoms (although their early development phases and/or museum preservation might have hindered pathological identification).

## PPC amplicon

## NAG01a-c amplicon

**Figure 1.** Region of the NAG01 Perkinsea SSU rRNA-encoding gene targeted by the qPCR assay. Sense-strand sequence of the region targeted for the PPC and NAG01a-c assays; the bold sequence is the variant region used to distinguish the strains. The red nucleotides are the sites that differ between the two strains, the arrows represent the forward and reverse primers, and the probe-like shapes represent the FAM (green) and Cy5 (red) probes.

**Table 1.** Primers and probes developed to target two clades of NAG01 Perkinsea: PPC and NAG01a-c. '_P1' indicates an oligonucleotide probe sequence. All primers and probes were manufactured by Eurofins Genomics and purified using high-performance liquid chromatography.

| primer/probe | sequence (5'-3') | specificity | reference |
|---|---|---|---|
| 300F-B | GGGCTTCAYAGTCTTGCAAT | NAG01 Clades A–D | [6] |
| NAG01R_1 | GCCTGCTTGAAACRCTCTAA | NAG01 Clades A–D | this study |
| PPC_P1 | FAM-TGCCAAGAACGACCGTCCTAC-BHQ1 | NAG01 Clade D (PPC) | this study |
| NAG01a-c_P1 | Cy5-CAAGGACGACCTACCCACCTTAG-BHQ2 | NAG01 Clades A–C (NAG01a-c) | this study |

## 2.3. Plasmid template DNA

### 2.3.1. Pathogenic Perkinsea clade assay

The PPC 18S rDNA region was amplified from KNA_DNA (above) using primers 300F-B/NAG01R_1 (table 1) and cloned into pSC-A-amp/kan using the StrataClone PCR Cloning Kit (Agilent Technologies) to generate pPPC (4590 bp). Plasmid DNA was extracted using the GeneJET Plasmid Miniprep Kit (Thermo Fisher), linearized using NcoI, purified using the GeneJET PCR Purification Kit (Thermo Fisher) and quantified using a Qubit dsDNA BR assay kit (Invitrogen).

### 2.3.2. NAG01a-c assay

The NAG01a-c 18S rDNA region was synthesized *de novo* and cloned into the pUC57-Amp plasmid vector (Synbio Technologies) to generate pNAG01a-c (2916 bp). This plasmid was purified and quantified, as above.

## 2.4. Quantitative PCR

qPCR reactions were performed on the CFX96 real-time PCR detection system (Bio-Rad). Each 25 µl reaction contained: 1X TaqMan Gene Expression Master Mix (Thermo Fisher), 500 nM of each primer, 250 nM of (each) probe, 1 µl of template DNA and 5% DMSO. The optimal qPCR conditions were determined to be 10 min at 95°C followed by 40 cycles of 15 s at 95°C and 2 min at 60°C (PPC), 62°C (NAG01a-c), or 61°C (duplex). All reactions were performed in at least triplicate, unless indicated otherwise, alongside a no-template control (NTC).

# 3. Results

## 3.1. Assay optimization

Both the PPC and NAG01a-c qPCR assays were tested at a six-point annealing temperature (Ta) gradient, ranging from 55.6°C to 64°C (figure 2), using 1 ng of pPPC or pNAG01a-c template. The PPC qPCR assay

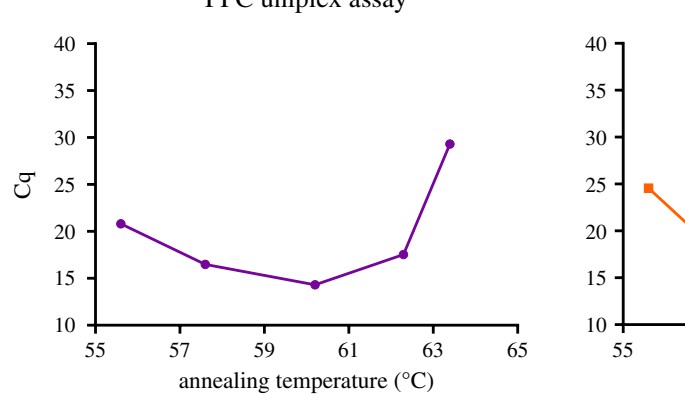
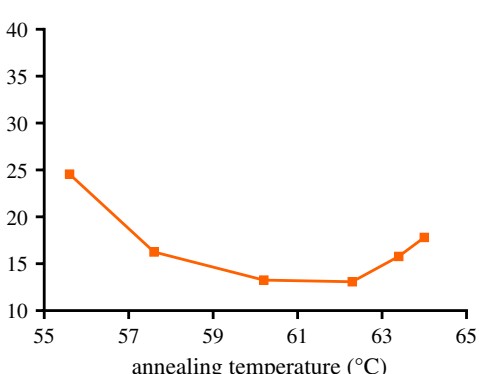

**Figure 2.** Quantitation cycle (Cq) plotted against a six-point temperature gradient. Plots were used to determine the optimal annealing temperatures (Ta) for each of the PPC and NAG01a-c uniplex assays. Each Ta was selected based on the temperature at which the earliest amplification (i.e. the lowest Cq value) was observed.

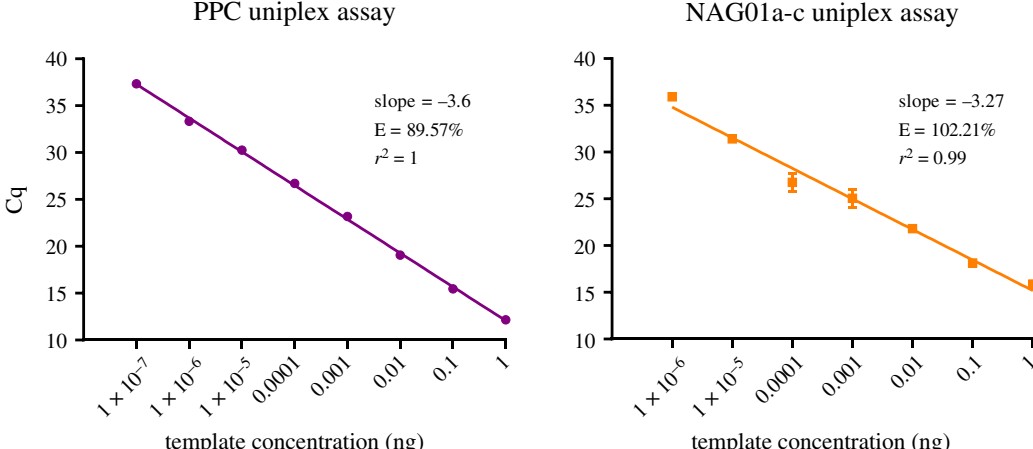

**Figure 3.** Standard curves for the PPC and NAG01a-c uniplex assays. Cq values plotted against a 10-fold plasmid template dilution series. The graphs depict the linear dynamic range, corresponding to the range of template concentrations in which the assay has high amplification efficiency (E = amplification efficiency). Error bars represent one standard deviation from the mean; these are small enough to be obscured behind each data point in the PPC uniplex assay.

successfully detected amplification at five Ta's ranging from 55.6°C to 63.4°C (figure 2); 60°C was retained as Ta for the PPC qPCR assay. The NAG01a-c qPCR assay successfully detected amplification at all six temperatures (figure 2); 62°C was retained as the Ta for the NAG01a-c qPCR assay. No fluorescent signal was detected in the NTCs for either assay.

## 3.2. Assay sensitivity

### 3.2.1. Pathogenic Perkinsea clade uniplex assay

The upper limit of detection (LoD) of the assay was determined to be $2.02 \times 10^8$ target copies, and the lower LoD was 202 target copies (figure 3); these values were determined as the maximum/minimum number of copies that can be accurately and reproducibly measured for each assay. The assay efficiency was 89.57%, and the linear dynamic range is shown in figure 3.

### 3.2.2. NAG01a-c uniplex assay

The upper LoD of the assay was $3.18 \times 10^8$ copies, while the lower LoD of the NAG01a-c assay corresponded to the lowest point of the linear dynamic range: 318 copies (figure 3). The assay efficiency was 102.21%; the linear dynamic range is also shown in figure 3.

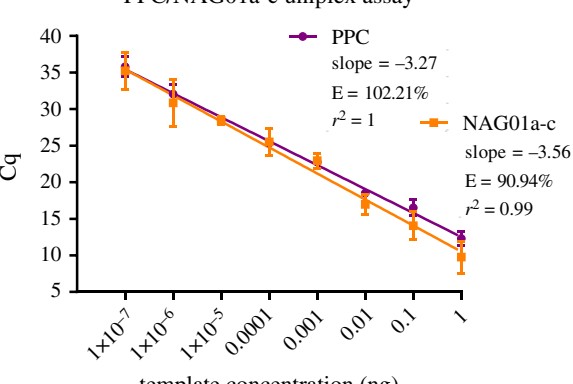

**Figure 4.** Standard curve of the PPC/NAG01a-c duplex qPCR assay. Cq values plotted against a 10-fold plasmid template dilution series. The graphs depict the linear dynamic range for each assay (E = amplification efficiency); error bars represent one standard deviation from the mean.

### 3.2.3. Pathogenic Perkinsea clade/NAG01a-c duplex assay

The overall amplification efficiencies for the PPC and NAG01a-c assays were slightly altered in duplex conditions, at 102.21% and 90.94%, respectively (figure 4). The upper LoDs for PPC and NAG01a-c were determined to be $2.02 \times 10^8$ copies and $3.18 \times 10^8$ copies, respectively, and the lower LoDs were determined to be 20 copies and 32 copies.

## 3.3. Assay specificity

Assay specificity was tested *in vitro* at each stage of development by including three or four qPCR reaction replicates using non-target (i.e. the Perkinsea clade not targeted by the qPCR probe) plasmid template (1 ng); there was no detection of pPPC with the NAG01a-c probe, and no detection of pNAG01a-c with the PPC probe, confirming that these probes are discriminatory.

## 3.4. Applying the assay to tissue samples

The assay was assessed using tadpole liver tissue from Alaska, USA (KNA_DNA; PPC) and from French Guiana (G8.1/G2.13; NAG01a), the latter previously sampled as part of an earlier study [6]. KNA_DNA (10 ng) was tested in duplicate to assess the performance of the PPC uniplex assay, yielding an average Cq of 20.23 (electronic supplementary material, table S2), which corresponds to approximately $1.06 \times 10^6$ target gene copies. The French Guiana extractions were tested to assess the performance of the NAG01a-c uniplex assay. These samples, collected in 2013, were much older than the Alaskan samples, and the nature of the infection load was lower. One of the samples (G8.1) amplified successfully, yielding an average Cq of 33.89 (electronic supplementary material, table S2), corresponding to approximately 75 target gene copies. G2.13 did not produce a detectable signal; possible reasons for this are discussed later. KNA_DNA and G8.1 were tested together in order to evaluate the performance of the duplex assay: PPC was detected successfully from KNA_DNA in the combined sample, yielding an average Cq of 21.41, similar to the result of the PPC uniplex assay (electronic supplementary material, table S2). However, NAG01a-c was not detected from G8.1 with the duplex assay when it was analysed in combination with KNA_DNA. By contrast, when the G8.1 genomic DNA was assayed with a low concentration ($1 \times 10^{-6}$ ng) of pPPC plasmid, G8.1 amplified successfully, yielding an average Cq of 32.89 across the three replicates, a similar value to that of the NAG01a-c uniplex assay. This indicates that at higher concentrations of PPC template, lower levels of NAG01a-c template are not detectable with this assay; this is discussed further below.

## 3.5. Applying the assay to a field-compatible qPCR platform

The qPCR assay was tested on a handheld, field-compatible qPCR device, the *Biomeme two3*™. The device is battery-powered and docks to an Apple iPhone 5S that runs a custom app for controlling the device's electronics [9]. The reaction mixture used with the *two3*™ was consistent with the previous assays performed on the CFX96 system, except that TaqMan Gene Expression Master Mix (Thermo

Fisher) was substituted with LyoDNA 2.0 Master Mix (Biomeme). Cycling conditions were the same on both qPCR platforms. The upper and lower LoDs of each uniplex assay and the duplex assay were tested using known concentrations of plasmid DNA. Subsequently, the uniplex and duplex assays were also tested using genomic DNA from KNA_DNA (PPC) and G8.1 (NAG01a).

### 3.5.1. Uniplex assay

The upper and lower LoDs for the PPC uniplex assay were detected successfully on the *two3*™, yielding Cq values of 10.93 and 30.54. KNA_DNA was detected with the PPC uniplex assay, yielding a Cq of 18.46. The upper and lower LoDs for the NAG01a-c uniplex assay were detected successfully on the *two3*™, yielding Cq values of 10.84 and 37.82. G8.1 was detected with the NAG01a-c uniplex assay, yielding a Cq of 32.67. There was no detection of pPPC with the NAG01a-c probe, and no detection of pNAG01a-c with the PPC probe, confirming that the probes are discriminatory.

### 3.5.2. Duplex assay

The upper and lower LoDs for PPC and NAG01a-c assays in duplex conditions were detected successfully, yielding Cq values of 11.47 and 37 (pPPC) and 13.66 and 38.25 (pNAG01a-c). Using DNA from tissue samples KNA_DNA and G8.1 combined together, and using the qPCR duplex conditions, PPC was detected from KNA_DNA with a Cq of 17.68; however, NAG01a-c again failed to be detected from G8.1 genomic DNA. However, when G8.1 genomic DNA was assayed with a lower concentration ($1 \times 10^{-6}$ ng) of pPPC plasmid, G8.1 was detected successfully, yielding a Cq of 34.2. Again, this indicates that at higher concentrations of PPC template, lower levels of NAG01a-c template are not detectable. Nevertheless, in the majority of cases, the results of the tests performed on the *two3*™ were similar to the results obtained on the CFX96 platform.

## 4. Discussion

Quantitative PCR is a sensitive tool for diagnosing infection and has been routinely applied to various amphibian populations, leading to a better understanding of host–pathogen dynamics [8,10–13]. We have developed a qPCR assay that can rapidly detect, and allow stepwise discrimination between, two clades that include an emerging alveolate pathogen of amphibians, associated with either symptomatic (PPC) or putatively asymptomatic (NAG01a-c) infections. We have shown that this assay can then be easily performed on a portable handheld qPCR instrument, with potential applications for in-the-field diagnostics.

Despite the lower LoD of our PPC uniplex assay being slightly higher than that of Karwacki *et al.* [8], the symptomatic tissue samples used in this study (KNA_DNA: electronic supplementary material, table S3) consistently amplified with the PPC assay (in both uniplex and duplex conditions and on both qPCR platforms), yielding relatively low Cq values. Indeed, symptomatic samples typically contain many Perkinsea organisms [1] and therefore a large amount of target DNA is likely to be available for detection. By contrast, putatively asymptomatic tissue samples are likely to contain fewer Perkinsea cells [6] and therefore less target DNA. Consequently, our results obtained from putatively asymptomatic samples (G2.13 and G8:1, both NAG01a+) are much more ambiguous. Indeed, only one of these samples (G8.1) amplified with the NAG01a-c assay (in both uniplex and duplex conditions and on both qPCR platforms), and the Cq values were relatively high. Based on these values, we determined that the G8.1 sample contained only approximately 75 target gene copies; this is towards the limit of detection of the NAG01a-c duplex assay (electronic supplementary material, table S2). We presume that either the amount of target DNA in G2.13 (the NAG01a+ sample that did not amplify with the assay) is less than the lower LoD, that the sample has degraded from the time when it was screened for NAG01 DNA by Chambouvet *et al.* [6], or that PCR inhibitors are present in the tadpole tissue that limit the detection of NAG01a. Given that G8.1 amplified successfully at a level of approximately 75 copies, we suggest that a concurrently low level of target DNA in G2.13 is the most likely scenario, meaning that some low-level NAG01a-c infections may be overlooked in this duplex assay.

### 4.1. Limitations

We cannot rule out that extremely low levels of NAG01a-c infection, or indeed early stage low-level PPC infections, may not be detectable, though, for sample G2.13, this may be due to sample degradation. It

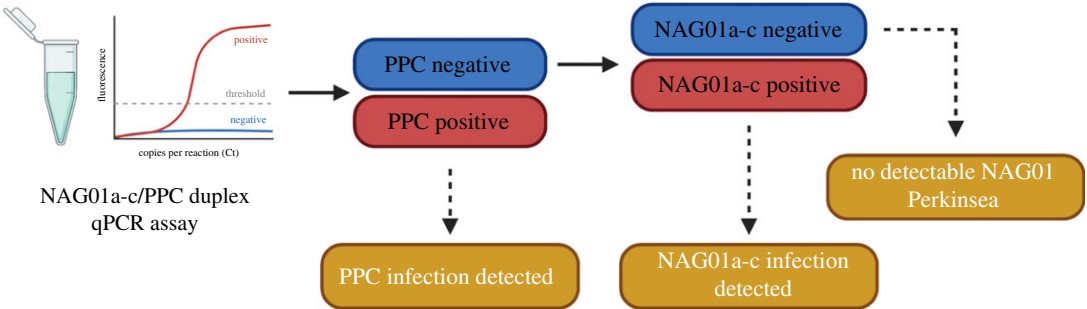

**Figure 5.** Stepwise assessment of PPC/NAG01a-c template presence/absence. The duplex assay can determine whether an abundant template, typically PPC, is present (positive) or absent (negative). If absent, the assay also allows for simultaneous detection of the other clade (typically the asymptomatic NAG01a-c clade). If neither assay is positive, this indicates that there is no detectable NAG01 Perkinsea infection. We note that it is likely that the stepwise detection assay will also operate in the opposite order, if the NAG01a-c Perkinsea infection is composed of abundant cells combined with a PPC infection of a low number of cells. Figure generated using Biorender.com.

may also be because individual liver samples may not always liberate sufficient genomic DNA to allow for NAG01 detection, particularly for low-level infections. However, this indicates that the limitation is in the level of sample material, rather than the detection method *per se*. If necessary, this could be overcome by the use of pooled samples to ensure sufficient input material for genomic DNA extractions.

The low amount of target DNA contained by putatively asymptomatic samples is important to consider when applying the duplex assay, as our results suggest that large amounts of PPC template can 'overwhelm' the signal emitted by the NAG01a-c template. This could potentially be a problem for e.g. PPC/NAG01a-c co-infected samples. For this reason, we propose this duplex assay as a means of rapid presence/absence detection for the two clades of amphibian Perkinsea, allowing stepwise assessment. Specifically, the assay allows for the investigation of a relatively high load of one Perkinsea, likely to be representatives of the PPC clade. If absent, the assay allows for secondary detection of a low-level infection, likely to be NAG01a-c, which is putatively an asymptomatic infection (figure 5). We advocate this stepwise detection strategy as ideal, because the aim of any such assay should be to primarily detect, or rule out, the presence of DNA from the PPC clade. Samples that are positive for either clade should then be amplified with both uniplex assays to get a more reliable estimate of template concentration.

# 5. Conclusion

We have shown that this novel duplex qPCR assay (i) can detect both PPC and NAG01a-c infections in genomic DNA extracted from tadpole tissue; (ii) can reliably discriminate between PPC and NAG01a-c infections in a stepwise manner (figure 5); and (iii) can be applied to a portable handheld instrument. Moving forward, we anticipate that this assay can be routinely applied to screening programmes to assess the prevalence of Perkinsea infections in both wild and captive tadpole populations, including captive breeding programmes for endangered species.

Ethics. Sampling in Alaska was carried out under FRP permit number SF2016–154. No invasive procedures were conducted on live tadpoles; these were euthanized using an overdose of MS-222 prior to dissection. The findings and conclusions in this article are those of the authors and do not necessarily represent the views of the U.S. Fish and Wildlife Service.

Data accessibility. The datasets supporting this article have been uploaded as part of the electronic supplementary material.

Authors' contributions. V.S. and D.S.M. designed the assay; A.C. and M.R. coordinated fieldwork and collected field samples; V.S. and D.S.M. conducted lab work, analysed data and drafted the manuscript; A.C. and T.A.R. critically reviewed the manuscript.

Competing interests. We have no competing interests.

Funding. This work was supported by Royal Society Challenge Grant (GCRF) 2017 'Assessing protist pathogen threats to endangered ecological keystone frog species of Panama' CHG\R1\170042 and a Horizon 2020 research and innovation award under the Marie Skłodowska-Curie ITN project SINGEK (http://www.singek.eu; grant agreement no. H2020-MSCA-ITN-2015-675752). T.A.R. is supported by a Royal Society University Research Fellowship (UF130382). A.C. was supported by the ANR project ACHN 2016 PARASED (ANR-16_ACHN_0003)

and by the French National programme EC2CO (Ecosphère Continentale et côtière) project Thrausto (grant no. 13046). The United States Fish and Wildlife Service, National Wildlife Refuges Program provided field housing and transportation and supported staff time for the research.

Acknowledgements. We thank Dr Anke Lange for her assistance with the CFX-96 real-time PCR detection system.

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
