## [Peer Review File · Royal Society Open Science]

Review History

RSOS-202150.R0 (Original submission)

Review form: Reviewer 1

Is the manuscript scientifically sound in its present form?

Yes

Are the interpretations and conclusions justified by the results?

Yes

Is the language acceptable?

Yes

Do you have any ethical concerns with this paper?

No

Have you any concerns about statistical analyses in this paper?

No

Recommendation?

Accept with minor revision (please list in comments)

Comments to the Author(s)

The manuscript presents a new (portable) assay that can be used in the field to detect two Perkinsea groups in amphibian tissues. The manuscript is well structured and clear and I would like to see it published. I only have a few minor comments that require clarification:

How were the primers designed?

Why did the authors pooled the liver tissue from each of the Alaska tadpoles prior to DNA extraction?

Please avoid terms like "assay was run"

Review form: Reviewer 2**Is the manuscript scientifically sound in its present form?**

Yes

Are the interpretations and conclusions justified by the results?

Yes

Is the language acceptable?

Yes

Do you have any ethical concerns with this paper?

No

Have you any concerns about statistical analyses in this paper?

No

Recommendation?

Accept as is

Comments to the Author(s)

First of all, I would like to apologize to the authors and the editors for the delay reviewing their paper.

The paper is well-written and clear, the methodologies sound, and the developed diagnostic strategy useful in a field setup. So, I am happy to recommend the publication of the manuscript as it is after deleting the word "library" from line 17 page 4 as I believe the authors mean just "clone".

Decision letter (RSOS-202150.R0)

Dear Dr Milner

On behalf of the Editors, we are pleased to inform you that your Manuscript RSOS-202150 "A novel duplex qPCR assay for stepwise detection of multiple *Perkinsea* protistan infections of amphibian tissues" has been accepted for publication in Royal Society Open Science subject to minor revision in accordance with the referees' reports. Please find the referees' comments along with any feedback from the Editors below my signature.

Please submit your revised manuscript and required files (see below) no later than 7 days from today's (ie 27-Jan-2021) date. Note: the ScholarOne system will 'lock' if submission of the revision is attempted 7 or more days after the deadline. If you do not think you will be able to meet this deadline please contact the editorial office immediately.

on behalf of Professor John Dalton (Associate Editor) and Pete Smith (Subject Editor)
openscience@royalsociety.org

Associate Editor Comments to Author (Professor John Dalton):

You manuscript reviewed very favourable reviews. One reviewer request consideration of two important questions -

How were the primers designed?

Why did the authors pooled the liver tissue from each of the Alaska tadpoles prior to DNA extraction?

Reviewer comments to Author:

Reviewer: 1

Comments to the Author(s)

The manuscript presents a new (portable) assay that can be used in the field to detect two *Perkinsea* groups in amphibian tissues. The manuscript is well structured and clear and I would like to see it published. I only have a few minor comments that require clarification:

How were the primers designed?

Why did the authors pool the liver tissue from each of the Alaska tadpoles prior to DNA extraction?

Please avoid terms like "assay was run"

Reviewer: 2

Comments to the Author(s)

First of all, I would like to apologize to the authors and the editors for the delay reviewing their paper.

The paper is well-written and clear, the methodologies sound, and the developed diagnostic strategy useful in a field setup. So, I am happy to recommend the publication of the manuscript as it is after deleting the word "library" from line 17 page 4 as I believe the authors mean just "clone".

===PREPARING YOUR MANUSCRIPT===

===PREPARING YOUR REVISION IN SCHOLARONE===

To revise your manuscript, log into <https://mc.manuscriptcentral.com/rsos> and enter your Author Centre - this may be accessed by clicking on "Author" in the dark toolbar at the top of the

page (just below the journal name). You will find your manuscript listed under "Manuscripts with Decisions". Under "Actions", click on "Create a Revision".

<https://royalsociety.org/journals/authors/author-guidelines/#supplementary-material> to include a suitable title and informative caption. An example of appropriate titling and captioning may be found at https://figshare.com/articles/Table_S2_from_Is_there_a_trade-off_between_peak_performance_and_performance_breadth_across_temperatures_for_aerobic_sc_ope_in_teleost_fishes_/3843624.

Author's Response to Decision Letter for (RSOS-202150.R0)

See Appendix A.

Decision letter (RSOS-202150.R1)

Dear Dr Milner,

It is a pleasure to accept your manuscript entitled "A novel duplex qPCR assay for stepwise detection of multiple Perkinsea protistan infections of amphibian tissues" in its current form for publication in Royal Society Open Science.

on behalf of Professor John Dalton (Associate Editor) and Pete Smith (Subject Editor)
openscience@royalsociety.org

Appendix A

UNIVERSITY OF OXFORD
DEPARTMENT OF ZOOLOGY
Zoology Research and Administration Building
11a Mansfield Rd, Oxford OX1 3SZ
e-mail: David.Milner@zoo.ox.ac.uk

From:
Dr David S. Milner
Research Fellow

3rd February 2021

Dear Editors and Reviewers,

Thank you for the careful consideration of our manuscript, and the positive comments. In response to the reviewers' requests, we enclose an updated manuscript with the following changes:

How were the primers designed?

Primer 300F-B was originally described by Chambouvet *et al.* 2015 (please see Table 1). Primer NAG01R_1 was designed to work in tandem with 300F-B; this primer and both probes were designed manually, based on the full alignment of *Perkinsea* sequences available at doi.org/10.5281/zenodo.12712. We have added the latter point to the Materials and Methods section.

Why did the authors pool the liver tissue from each of the Alaska tadpoles prior to DNA extraction?

The liver tissue from the Alaska tadpoles was pooled to ensure a sufficient amount of starting material for the DNA extraction protocol, as the individual samples were very small. We mention this as a limitation of the assay in the *Limitations* section of the *Discussion* and have also added this point to the Materials and Methods section.

Please avoid terms like "assay was run"

We have removed all instances of this in the manuscript and replaced with either 'tested', 'analysed' or 'performed'

The paper is well-written and clear, the methodologies sound, and the developed diagnostic strategy useful in a field setup. So, I am happy to recommend the publication of the manuscript as it is after deleting the word "library" from line 17 page 4 as I believe the authors mean just "clone".

Thank you for these positive comments. We have removed this instance of 'library'

Yours sincerely,

David Milner